# First Trimester Ultrasound Detection of Fetal Central Nervous System Anomalies

**DOI:** 10.3390/brainsci13010118

**Published:** 2023-01-09

**Authors:** Delia Roxana Ungureanu, Roxana Cristina Drăgușin, Răzvan Grigoraș Căpitănescu, Lucian Zorilă, Anca Maria Istrate Ofițeru, Cristian Marinaș, Ciprian Laurențiu Pătru, Alexandru Cristian Comănescu, Maria Cristina Comănescu, Ovidiu Costinel Sîrbu, Maria-Sidonia Vrabie, Lorena Anda Dijmărescu, Ioana Streață, Florin Burada, Mihai Ioana, Alice Nicoleta Drăgoescu, Dominic Gabriel Iliescu

**Affiliations:** 1Doctoral School, University of Medicine and Pharmacy of Craiova, 200349 Craiova, Romania; 2Department of Obstetrics and Gynecology, University of Medicine and Pharmacy of Craiova, 200349 Craiova, Romania; 3Department of Obstetrics and Gynecology, Emergency Clinical County Hospital, 200642 Craiova, Romania; 4Department of Histology, University of Medicine and Pharmacy of Craiova, 200349 Craiova, Romania; 5Department of Human Anatomy, University of Medicine and Pharmacy of Craiova, 200349 Craiova, Romania; 6Department of Obstetrics and Gynecology, Filantropia County Hospital, 200143 Craiova, Romania; 7Laboratory of Human Genomics, University of Medicine and Pharmacy of Craiova, 200638 Craiova, Romania; 8Regional Centre of Medical Genetics Dolj, Emergency Clinical County Hospital Craiova, 200642 Craiova, Romania; 9Department of Anesthesia and Intensive Care, Emergency Clinical County Hospital, 200642 Craiova, Romania; 10Department of Anesthesia and Intensive Care, University of Medicine and Pharmacy of Craiova, 200349 Craiova, Romania

**Keywords:** central nervous system, first trimester, prenatal diagnosis, ultrasound, anomaly scan, spina bifida, holoprosencephaly

## Abstract

Objective: To evaluate the potential of the first-trimester ultrasound (US) features for the detection of central nervous system (CNS) anomalies. Methods/Methodology: This is a prospective one-center three-year study. Unselected singleton pregnant women were examined using an extended first-trimester anomaly scan (FTAS) that included the CNS assessment: the calvaria shape, the septum (falx cerebri), the aspect of the lateral ventricles, the presence of the third ventricle and aqueduct of Sylvius (AS) and the posterior brain morphometry: the fourth ventricle, namely intracranial translucency (IT), brain stem/brain stem–occipital bone ratio (BS/BSOB) and cisterna magna (CM). The spine and underlying skin were also evaluated. The cases were also followed during the second and third trimesters of pregnancy and at delivery. FTAS efficiency to detect major CNS abnormalities was calculated. Results: We detected 17 cases with CNS major abnormalities in a population of 1943 first-trimester (FT) fetuses, including spina bifida with myelomeningocele, exencephaly-anencephaly, holoprosencephaly, hydrocephaly, cephalocele and Dandy-Walker malformation. The CNS features in the abnormal group are presented. In the second trimester (ST), we further diagnosed cases of corpus callosum agenesis, cerebellar hypoplasia, vein of Galen aneurysm and fetal infection features (ventriculomegaly, intraventricular bands, intraventricular cyst and hyperechoic foci), all declared normal at the FTAS. During the third trimester (TT) scan we identified a massive fetal cerebral haemorrhage absent at previous investigations. We report a detection rate of 72.7% of fetal brain anomalies in the FT using the proposed CNS parameters. The sensitivity of the examination protocol was 72.7%, and the specificity was 100%. Conclusion: A detailed FT CNS scan is feasible and efficient. The majority of cases of major CNS abnormalities can be detected early in pregnancy. The visualization rates of the CNS parameters in the FT are great with short, if any, additional investigation time. FT cerebral disorders such as haemorrhage or infections were missed in the FT even when an extended evaluation protocol was used.

## 1. Introduction

The FTAS routine assessment has two primary objectives in obstetric practice: evaluation of sonographic genetic markers that increases the accuracy and sensitivity of detecting chromosomal abnormalities and early detection of congenital anomalies [1,2,3].

FT evaluation of the CNS is challenging as the cerebral structures evolve considerably and continuously change in shape and form during pregnancy [4,5]. Still, the fetal brain is the first organ system to develop in such a way that it can be imaged early in pregnancy by transvaginal US [6].

The assessment and early detection of CNS malformations is critical as they represent the largest group of fetal abnormalities, with a prevalence of more than 10 cases in 1000 births, more than Down syndrome and similar to congenital cardiac diseases [7,8]. Acrania, exencephaly, anencephaly, holoprosencephaly, cephalocele and spina bifida represent major abnormalities with significant prevalence, limited therapeutic solutions and the worst short and long-term prognosis that can be easily diagnosed before 14 weeks [9,10,11].

Early detection of major CNS diseases offers the possibility for a safer termination of pregnancy with less emotional stress and less economic costs [12].

This article aims to evaluate the role of a detailed FTAS of the fetal head and spine to improve the detection rate of major fetal CNS anomalies.

## 2. Materials and Methods

This prospective research was performed in two referral centers, the Prenatal Diagnostic Unit of the County Emergency Hospital of Craiova and Ginecho Clinic Craiova from October 2019 until October 2022. We included 1943 singleton pregnancies evaluated for the 12–13 weeks + 6 days FTAS. We excluded all multiple pregnancies, non-viable pregnancies and all patients lost to follow-up. All eligible patients had to sign an informed consent for participation and use of the acquired data during the study period. The study was approved by the Ethics Committee of the University of Medicine and Pharmacy Craiova (No 29/20.03.2018).

In terms of FTAS, we adopted a standard protocol that added the guidelines for basic FT screening [8] and several features for a more detailed structural assessment at this gestational age. The detailed scanning protocol included two axial planes of the brain (at the level of the choroid plexus and third ventricle and the level of the cerebral peduncles and aqueduct of Sylvius) and the sagittal view of the fetal face for the evaluation of the posterior brain complex. The scanning protocol aimed to assess the contour and shape of the fetal skull, the choroid plexus shape and relative filling of the ventricles and the aspect of the lateral cerebral ventricles (Figure 1A). We noted the presence of the third ventricle (Figure 1B) and the aspect and position of AS (Figure 1C), the presence of the CM and the fourth ventricle (the anechogenic area between the posterior border of the brain stem (BS) anteriorly and the choroid plexus of the fourth ventricle posteriorly) (IT) [13]. We investigated the presence of posterior and caudal displacement of the mesencephalon by searching for the ‘crash’ sign in the thalamic axial plane [14]. The BS/BSOB ratio was subjectively evaluated and measured when it appeared abnormal [15], and the presence of CM was noted (Figure 1D). Fronto-maxillary angle was measured in all cases, as part of the genetic screening, but narrow-angle cases were suspected also for neural tube defects [16]. In all cases, the spine was also evaluated in two-dimensional (2D) longitudinal planes, recording its regularity and the continuity of the underlying skin layer (Figure 1E).

The sonographers were invited to record all clinical situations that impaired the proper transabdominal visualization of the first trimester CNS features presented above. In these cases, reevaluation or transvaginal scan was proposed for the study group.

The ST anomaly scan is essential as it represents the most important fetal morphological assessment. It is being offered in most states as a screening method for fetal malformation detection between 18 to 22 gestational weeks [17]. In the second and third trimesters, we evaluated the fetal head and spine in 5 planes: the trans-ventricular plane, the trans-cerebellar plane, the trans-thalamic plane, the longitudinal plane of the spine and the mid-sagittal or median plane for assessing all major midline organs and their anomalies. As recommended, we also noted the intact cranium, the cavum septi pellucidi, the midline falx, the thalami, the cerebral ventricles, the cerebellum and the cisterna magna and also if there was no spinal defects or masses (in transverse and sagittal views).

All scans were performed using Voluson E8 and E10 systems (GE Healthcare) equipped with transducers for transabdominal (3–9 MHz) and transvaginal (5–9 MHz) examinations. In the first trimester, the transabdominal approach was preferred in most cases, while transvaginal sonography was carried out only in selected cases to obtain good visualization of the fetal CNS anatomy.

The Department of Medical Genetics of the University of Medicine and Pharmacy Craiova provided genetic counsel and performed genetic tests in all requested cases of postabortum specimens.

After termination of pregnancy, an autopsy of the fetuses with major FT CNS anomalies was offered and performed in the Pathology Department of the County Emergency Hospital of Craiova as an audit for the US diagnosis. We noted the gestational age at diagnosis, the genetics and the outcome of the abnormal cases.

The study aimed to present the US features that helped diagnose CNS abnormalities in the FT.

All data were analyzed to establish the efficiency of the detailed CNS protocol in detecting major CNS anomalies in the FT using the statistical software of Microsoft Office Excel packages.

## 3. Results

The study included 1943 pregnancies examined in the FT using the proposed CNS examination protocol. We report 17 cases of major FT CNS anomalies, of which 2 were lost to follow-up. We diagnosed three cases of open spina bifida (OSB), one case of close spina bifida (CSB), three cases of anencephaly, three cases of holoprosencephaly, one case of ventriculomegaly, two cases of cephalocele and two cases of Dandy-Walker malformation. One thousand five hundred and sixty-four cases declared normal in the FT were monitored in the second and third trimesters of pregnancy and 354 cases were lost to follow-up. In the ST, we diagnosed two cases of corpus callosum agenesis and one case of corpus callosum agenesis associated with Dandy-Walker malformation, two cases of cerebellar hypoplasia and three cases of fetal infection. In the TT, we diagnosed one case of massive intracranial haemorrhage and one case of vein of Galen aneurysm. At the FT CNS evaluation, we noted isolated abnormal features in nine cases that were later declared normal in the ST: two cases of apparent ventriculomegaly, three cases with no visualization of the third ventricle and three cases with enlarged choroid plexus area giving the dry brain sign, one case of “crash” sign. The results are summarized in the flowchart (Figure 2).

The statistical analysis yielded a 99.4% accuracy in detecting fetal brain anomalies in the FT using the proposed CNS parameters. The sensitivity of the examination protocol was 72.7%, and the specificity was 100%. Our study declared a positive predictive value of 100% and a negative predictive value of 99.4%. The first-trimester detection rate for major CNS anomalies was 72.7%.

The transabdominal (TA) approach for the FT scan was preferred. All sonographers participating in the study declared a good visualization rate of all CNS features in approximately 93% of the cases using TAUS. Reevaluation and/or transvaginal (TV) approach were required in 7.05% of cases (Table 1) to complete the entire fetal CNS ultrasound protocol because of fibroids of the pregnant uterus (2.9%), retroverted uterus (8%), body mass index > 24 (23.3%), abdominal scar (16.7%), unfavorable fetal position (8%) and associations (34.8%) (Table 1).

In Table 2 we summarize the features of the 15 cases of FT CNS anomalies.

### 3.1. Spina Bifida

In the axial plane, we noted a decreased intracranial fluid with enlarged choroid plexus in one of three OSB cases while the ‘crash’ sign was obvious in two/three cases. Similarly, in two/three cases of OSB, we noted an abnormal posterior brain (IT, CM and BS/BSOB ratio) (Figure 3 and Figure 4). The single case of meningocele with CSB presented a normal posterior brain anatomy in axial and sagittal view [18]. We found abnormal features of the spine and the overlying skin (kyphoscoliosis, spinal defect or meningocele) in all cases of SB (4/4 cases).

### 3.2. Cephalocele

The study reports two cases of cephalocele: one located in the occipital region (Figure 5) and one located in the parietal region of the fetal head. Both cases presented with an abnormal calvaria showing the cranial bone defect with a herniated fluid-filled cyst.

### 3.3. Dandy-Walker Malformation

US assessment of the fossa posterior of the fetal brain allowed us to detect two cases of Dandy-Walker malformation with abnormal posterior brain morphometry: an increased fourth ventricle, CM and BSOB diameter. One case associated with hydrocephaly.

### 3.4. Acrania, Exencephaly, Anencephaly

All three cases of anencephaly were easily identified due to absence of the skull ossification and the clear deformity of the brain with no visualization of the normal CNS US features (Figure 6). The spine with the overlying skin was found unaffected.

### 3.5. Holoprosencephaly

Cerebral parenchyma, ventricular system and midline structures were strongly affected in all three cases of holoprosencephaly detected in the FT. A fusion of the anterior horns of the lateral ventricles and the absence of the butterfly sign of the fetal brain was described in all cases diagnosing alobar holoprosencephaly. One case of alobar holoprosencephaly is also associated with proboscis and extreme hypotelorism–synophthalmia (Figure 7).

### 3.6. Ventriculomegaly/Hydrocephaly

The single case of hydrocephaly was diagnosed due to enlarged lateral ventricles relative to the choroid plexus and increased third ventricle, IT and CM. We noted an abnormal sagittal view with an important disproportion between the head and the rest of the body (Figure 8).

Invasive genetic testing was proposed in all cases with abnormal CNS features, but only 60% (9/15 cases) of couples agreed to the test. However, it should be noted that 40% (6/15 cases) of the pregnant women diagnosed with CNS abnormalities declined invasive genetic testing. Chromosomal abnormalities accounted for 55.5% (5/9 cases) of cases with major CNS disorders. Medical termination of pregnancy (TOP) was proposed in all diagnosed cases to obtain a specimen for the pathology confirmation. 13.3% of couples (2/15 cases) preferred a surgical TOP and in 6.6% of cases (1/15 cases) there was a miscarriage. US prenatal diagnosis was later confirmed by the pathology exam in 80% of cases (12/15 cases) (open and closed spina bifida, exencephaly-anencephaly, holoprosencephaly, hydrocephaly and cephalocele).

## 4. Discussion

Ultrasonography is the primary screening modality for fetal imaging because of its cost-effectiveness and safety. It is widely used for assessing the fetal brain, including cerebral parenchyma, ventricular system and midline structures [19]. In this study, we report a detection rate of 72.7%, slightly higher than some recently published data (68.6% [20], 53%) [21]). This high accuracy for fetal CNS anomaly detection in the FT has been demonstrated as feasible by others [22,23]. The early scan detected all cases of acrania, alobar holoprosencephaly and cephalocele, as previously reported [24]. Still, the detailed CNS examination protocol enabled us to detect 100% of cases of SB, while a basic standard examination was reported with only a 14% detection rate at 11–13 weeks of gestation [25]. Corpus callosum agenesis and cerebellar hypoplasia were unmarked in the FTAS as these anomalies are considered undetectable early in pregnancy [25]. Ventriculomegaly, porencephaly, schizencephaly and neural migration defects secondary to congenital infection or brain haemorrhage usually manifest later in pregnancy.

Good visualization of all CNS features by the TA approach was obtained in 93%, while in 7% of cases, the sonographers need to reschedule or/and TV approach to obtain more information. We declare that the main conditions that hindered the proper assessment of the FT CNS features were maternal obesity and unfavorable fetal position. Our data are similar to others [26].

In this study, 50% of cases (4/8 of cases) with major CNS disorders presented a chromosomal anomaly. The non-visualization of the choroid plexus and the fourth ventricle in the FT has been reported to be associated with a high rate (71%) of genetic defects [27], while ventriculomegaly has been reported to be associated with only 6.3% incidence of genetic alterations [28]. All major CNS anomalies diagnosed early in the FT have an extremely severe prognosis and TOP was the preferred option for 93.3% (14/15 cases) of couples as studies have shown that complications are less common in the FT [29]. As mentioned before [30,31], we would like to underline the importance of perinatal autopsy as our study confirmed the US diagnosis in all 73% of the cases performed.

### 4.1. Spina Bifida

Spina bifida (SB) is caused by a failure in the closure of the neural tube, which normally occurs by the sixth week of gestation [3]. Improved ultrasound technology and increasing interest to detect neural tube defects made SB early prenatal diagnosis possible. We report an excellent diagnostic accuracy (100%) in detecting SB in the FT in an unselected population, while other published results of large studies vary between 40 to 80% of the detection rate [32]. In OSB, skin coverage is absent at the lesion site, consequently exposing the nervous tissue and meninges to the amniotic environment [33,34,35]. In CSB, there is no cerebrospinal fluid loss because of the defect’s full skin coverage [33,34,35]. These two main entities have very different prognostic implications and necessitate a clear distinction. Previous reports noted that the absence of indirect cerebral signs is the most valuable sonographic clue for differentiating the two entities [33]. However, in our experience, these findings were not clear-cut, as one-third of the OSB cases (33%) presented with normal posterior brain anatomy [36]. FT detection of SB provides parents more time to evaluate the option for intrauterine surgery [37] or, otherwise, the opportunity for early termination of pregnancy. In this study, TOP was the consented choice of the couples in 100% of cases of SB (4/4 cases), while in some other settings, with better facilities for SB fetal surgery, pregnancy termination is preferred only in 80% of cases [38].

### 4.2. Open Spina Bifida

In the 1980s, the prenatal diagnosis of OSB proved to be accurate by the description of the intracranial signs in the ST, such as the “lemon sign” (scalloping of the frontal bones) and the “banana” sign (caudal displacement of the cerebellum [39,40]. A FT screening study published in the 90s reported missing all 29 cases of OSB [41]. Advances in US technology made it possible to look directly for the spine defect in the FT [4]. However, diagnosing OSB by directly visualizing the spine lesions can be more challenging in the FT than in later cases. Around 50% of OSB can be overlooked, especially if the defect is low and accompanied by a small meningocele [42]. Our experience does not fully support this conclusion, as 100% of OSB cases were identified by a detailed examination of the spinal signs.

In the last decades, significant efforts have been directed towards FT US diagnosis of the open spinal defect using cranial and cerebellar markers to increase the detection rate [13,39,43]. In 33.3% of fetuses (1/3 cases), we observed decreased intracranial fluid spaces with an appearance of enlarged choroid plexus described as the ‘dried up brain’ sign in the FT [44]. Our examination protocol included the subjective evaluation of the midbrain relative to the occiput (‘crash sign’) and not the measurement of the distance from the AS to the occipital region. This novel US marker first described in 2019 [13], was absent in all declared normal fetuses and was detected in two cases of OSB together with the absence of a normal AS image, indicating a detection rate of 66.6% (two/three cases). These results are not entirely in agreement with other published data that report a higher detection rate from 85.7% [45] to 90.6% [14] of diagnosing OSB in the FT by using the “crash sign”. Other techniques for screening for OSB in the axial plane include measuring the BPD as fetuses with OSB have a BPD inferior to the fifth percentile (detection rate of 45–55%) [4,46] and a ratio between the BPD and the transverse abdominal diameter less than 1 (detection rate of 70%) [47]. This study did not follow the assessment of the BPD measurements, due to the time loss and low sensitivity and specificity. The third ventricle was absent in 66.6% of cases of OSB (2/3 cases) due to the cerebrospinal fluid leakage specific to this type of anomaly. All the suspected cases were also evaluated by transvaginal approach to obtain a better assessment of the brain features, as suggested by the literature [48].

In the FT mid-sagittal plane of the fetal head, when measuring the nuchal translucency and assessing the ossification of the nasal bone and fronto-maxillary facial angle [1], some CNS secondary features of OSB are apparent [13]. No additional time is needed for OSB indirect markers assessment because nuchal translucency measurement and nasal bone evaluation are mandatory at this gestational age. Sonographic screening markers of the posterior brain in OSB include a non-measurable, a decreased or absent CM, a thickened BS, and an increased BS/BSOB ratio of more than 1 [1,49,50]. In this study, 66.6% of cases of OSB (two/three cases) associated with absent IT. Firstly described by Chaoui et al., IT has been proposed as a screening marker for OSB in the low-risk population [51]. However, extensive subsequent studies found highly variable FT detection rates of OSB by using IT from 33.3% [52], 53.5% [53,54] to 100% [55,56]. Non-visualization or a decreased CM was noted in 66.6% of cases (two/three) while previous papers demonstrated a sensitivity for OSB of 50 to 73% [4,45,49,51,57]. In relatively small studies, an obliterated CM has been proposed as the most sensitive marker [55], with the best screening performance in detecting OSB in the FT [58,59]. In our study, 66.6% of cases of OSB presented an increased BS diameter and BS/BSOB ratio. Detection rates of these indirect signs have been reported to vary from 36% [60] to 100% [15] in a limited population.

In our experience, the CNS US markers of the mid-sagittal and axial planes are easily visualized and measured with just a short addition of time. When combining the above-mentioned intracranial and spinal signs, we report a 100% effectiveness of FT screening for OSB. Previously, a large retrospective study reported an improved detection rate of OSB in the FT from 15 to 53% [61]. Our much better results may be attributed to a small number of isolated OSB, but we will continue to evaluate our data prospectively.

### 4.3. Closed Spina Bifida

CSB has a better prognosis than OSB due to a much lesser involvement of the neural cord and the lack of hydrocephalus later in pregnancy [33]. Infants usually present with minimal neurologic symptoms and have a good outcome [61]. The actual incidence of CSB has not yet been established, as most published literature does not clearly separate OSB from CSB. Current data found a variable incidence of CSB from 7% [33] to 10% [62]. Our research noted a 25% incidence of CSB in all cases of SB and a 0.05% incidence in the general population. The FT detection rate for CSB was 100% (1/1) higher than the results shown by Liao et al. (28.5%) [52]. We present a case of CSB with large skin-covered myelomeningocele diagnosed only by direct examination of the spine. The downward displacement of the hindbrain is not present, and CSB could have been missed just by examining the posterior brain anatomy. In such cases, all FT CNS US markers (IT, AS, CM, BS/BSOB ratio) were within normal limits, as previously mentioned [18,61]. Ghi et al. found that the absence of FT intracranial signs is specific for CSB, but Liao et al. demonstrated a 29% rate of abnormal CNS US markers in cases of CSB [33,52].

### 4.4. Cephalocele

Cephalocele is a relatively rare neural tube defect described as a cranial bony defect with the protrusion of the brain tissue (also called encephalocele) in 63% of cases or of the meninges (also called meningoceles) in 37% of the affected pregnancies [63]. An early prenatal diagnosis of cephalocele is feasible by viewing the cystic cranial lesion in continuity with the brain and the enlarged rhombencephalic cavity [64]. In our study, the absence of one of the three posterior brain spaces in the sagittal view was present in all cases of cephalocele. In most cases, the defect was in the occipital region, as previously stated by the literature as the preferred location [63]. Some studies report associated fetal malformations in 65% of cases [4], while others found cephalocele as an isolated anomaly diagnosed by the FTAS in 66% of cases [63]. In our experience, we found no other associated fetal defects in 100% of cases (2/2 cases). The largest series of fetal cephalocele diagnosed in the FT found no cases of aneuploidy [63], but we confirmed Down syndrome in one of the two detected cases. The couples opted for pregnancy termination in all cases of cephalocele before the ST because of the poor prognosis secondary to the brain deformities present at the FTAS.

### 4.5. Dandy-Walker Malformation

DWM was usually diagnosed on a ST US examination, still, FT diagnosis has been reported [65,66]. In this study, we detected two cases of Dandy-Walker malformation (DWM) in the FT by examining the posterior fossa structures: BS, IT and CM in the same mid-sagital view that we used for the measurement of nuchal translucency. Differential diagnosis should include other cystic malformations such as mega cisterna magna, cerebellar hypoplasia, Blake’s pouch cyst and an arachnoid cyst [67]. One of the two cases (50% of fetuses) presented hydrocephaly. These findings are consistent with already published data that ventriculomegaly and encephalocele are the most common CNS anomalies associated with DWM [68]. While the prevalence of chromosomal abnormalities in fetuses with isolated DWM has been reported to be 16.3% [69], we report a 50% (1/2 cases) rate of chromosomal disorders, probably because of associated findings. The long-term prognosis of fetuses with DWM is still unknown and it depends on the presence or absence of associated malformations and genetic disorders [70]. Studies have demonstrated that the overall rate of several neurodevelopmental problems is 58% in fetuses with posterior fossa abnormalities, but normal karyotype and no other associated anomalies [69]. In our study, in both cases of DWM (100%), the couples opted for medical TOP.

### 4.6. Acrania, Exencephaly, Anencephaly

Acrania or exencephaly represents the absence of the cranial bones, with only a thin layer, if any, covering the brain. The persistent contact of the cerebral hemispheres with the neurotoxic amniotic fluid determines the progression of exencephaly to anencephaly, the end stage of this progressive destruction of cerebral tissue [6]. The Mickey Mouse appearance of the two split cerebral hemispheres dangling freely in the amniotic fluid was first described more than two decades ago [71]. Due to this specific US image present after 11 weeks of gestation, we report a detection rate of 100% of acrania (3/3 cases) in the FT, results compatible with previous studies [72]. The CNS US features evaluated by the study protocol presented a severe alteration in all cases. In our study, all three cases of anencephaly had no other fetal anomalies diagnosed, while data from other papers found associated fetal anomalies in 12% of cases [73]. Genetic testing was offered in all cases but performed only in one (normal karyotype). The reported rate of chromosomal disorders is globally low, approximately 1.8% (1 in 55 cases) [74]. Due to the poor prognosis, FT TOP was offered and carried out in two cases, while one case miscarried.

### 4.7. Holoprosencephaly

Holoprosencephaly is a severe CNS anomaly detected early in pregnancy in 0.4% of all cases [74]. The US features include a midline (falx) defect, accompanied by a large fluid collection and the fusion of the thalami [1]. This study reports a detection rate of 100% (3/3 cases) of alobar holoprosencephaly. The so-called butterfly sign (absence of visualization of both choroid plexus in the axial plane) was present in all cases, strengthening previous studies’ results that confirmed very high sensitivity and specificity of an early diagnosis [75,76]. Facial abnormalities such as facial asymmetry, hypotelorism, central clefts, and abnormal orbits can be associated in most cases [77]. We found facial anomalies in 33.3% of the cases: proboscis and extreme hypotelorism–synophthalmia. We detected chromosomal abnormalities, namely trisomy 18, in 33% of cases (1/3), but previous studies found a higher rate (66%), mainly trisomy 13 and 18 [78]. Lobar or semilobar holoprosencephaly were not present in our study group, but the FT examination usually misses them as they are more subtle to confirm [4]. A 3D “inversion rendering” algorithm to detect abnormal brain cavities in FT fetuses was proposed in 2008 by Kim et al. [79]. The examination proved feasible, very illustrative and helpful for a clear description of the early fetal brain [80]. The present study did not include a 3D assessment of the FT fetal brain because of the required supplementary skills and increased examination time. Most fetuses with alobar holoprosencephaly die shortly after delivery or have severe mental sequels and TOP is usually offered in all diagnosed cases. In our group, medical TOP was performed in 2 cases and one couple opted for surgical TOP. A fetal autopsy was possible in one case and confirmed the prenatal diagnosis of holoprosencephaly, including facial abnormalities.

### 4.8. Ventriculomegaly/Hydrocephaly

Because of its rare occurrence in early pregnancy, ventriculomegaly is not well defined in the FT. Thus, detection rates are reported to be up to 16% [48,81]. Routinely, the diagnosis of ventriculomegaly is subjectively made by an experienced fetal medicine specialist. In normal pregnancies, the ratio between choroid plexus and lateral ventricle diameter, length and area, decreases with fetal BPD showing a rather shrinkage of the choroid plexus than an increased lateral ventricle [48]. FT ventriculomegaly is often associated with aneuploidy, yet in cases of trisomy 13 and 18, there is a smaller ratio of choroid plexus to lateral ventricles area, while in cases of trisomy 21, the ratio remains the same [82]. Counseling cases of ventriculomegaly early in pregnancy is almost impossible as causes may vary from a brain development defect to chromosomal abnormalities or signs of haemorrhage or infection [83]. Rare cases of aqueduct stenosis can be noted in the FT associated with severe ventriculomegaly [4]. Therefore, fetal neuro-sonography should be mandatory if there is a suspicion of isolated ventriculomegaly in the FT [48]. In this study, we report one case of clear hydrocephaly at 13 weeks and 5 days of gestation that presented with an image of small choroid plexuses floating in the dilated cerebral ventricles. The sagittal view showed a significant disproportion between the fetal head and the rest of the body with evident frontal bossing. We proposed genetic testing that confirmed trisomy 21 and the couple opted for TOP at the beginning of the ST.

The originality of our study consists of using a detailed ultrasound protocol for the CNS evaluation in the first trimester. The FT detection rate of major CNS abnormalities using such a scanning protocol has not been analyzed yet, although previous research suggested the importance of cerebral early markers to detect a broader spectrum of CNS major abnormalities [55].

The limitation of our study is due to the low prevalence of CNS anomalies and the extension of the studied population. Thus, the reliability and accuracy of the early US scan could not be calculated for each type of major malformation. Further larger studies are needed for these statistical issues, but our research serves as a reliable protocol that involves few supplementary resources (examination time, training of the sonographers).

## 5. Conclusions

Our experience showed that a detailed evaluation of the CNS early in pregnancy is feasible and has a great potential to effectively detect major CNS anomalies, with an accuracy of 99.4%. Detecting CNS malformations in the FT allows for a safer and less expensive termination of pregnancy. In addition, earlier recognition of the normal anatomy of the fetus aids in decreasing patient anxiety.

This study also confirmed previous findings that combined intracranial signs such as obliterated CM, a BS/BSOB ratio greater than 1, and the presence of „crash” signs are very effective in detecting OSB in the FT. The direct OSB features (spinal defect and meningocele) are rarely present in the FT, and indirect features expression is variable, so we encourage the sonographers to assess all the intracranial signs described in routine midsagittal and axial planes (NT, choroid plexus and transthalamic BPD planes). Healthcare providers should be aware that FTAS should not be regarded as a replacement for the second or third-trimester evaluation as the brain grows and differentiates significantly throughout gestation. Many diseases such as gyration anomalies, tumors, mild ventriculomegaly, porencephaly, schizencephaly, haemorrhage or infectious sequels cannot be diagnosed until late in pregnancy.

## Figures and Tables

**Figure 1 brainsci-13-00118-f001:**
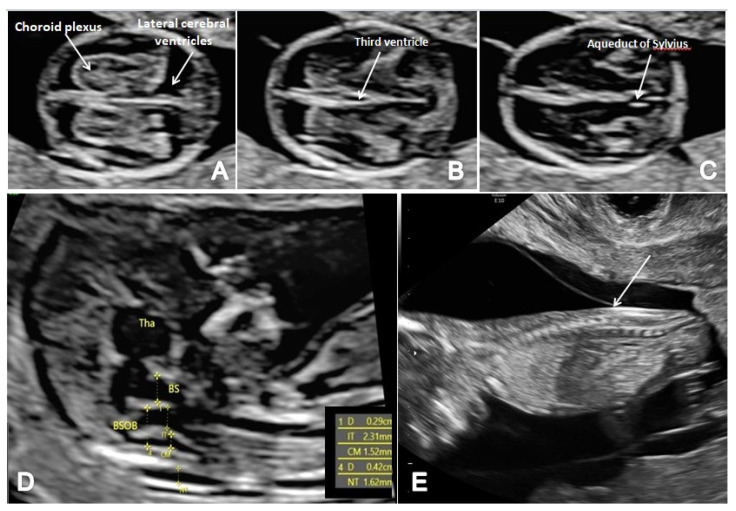
The FTAS protocol for CNS evaluation: (**A**) transverse view of the brain showing the contour and shape of the fetal skull, choroid plexus (arrow) and the filling of lateral cerebral ventricles (arrow); (**B**,**C**) further evaluation of the cerebral ventricular system, in transverse planes of the brain showing the third ventricle (**B**) and aqueduct of Sylvius (**C**) (arrow); (**D**) mid-sagittal view of the brain showing the thalamus (Tha) and the measurements for the brain stem (BS), the fourth ventricle (IT), cisterna magna (CM), the nuchal translucency (NT) and the brain stem–occipital bone ratio (BSOB); (**E**) longitudinal view of the spine regularity and underlying skin (arrow).

**Figure 2 brainsci-13-00118-f002:**
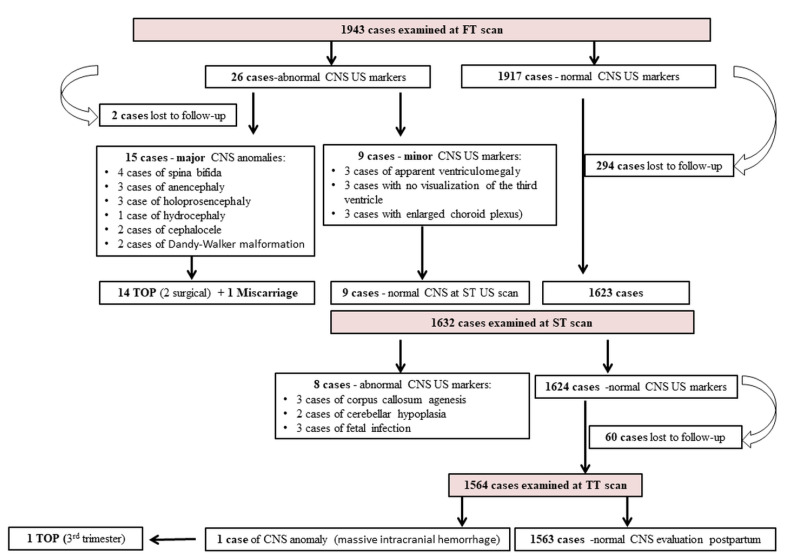
Flowchart summarizing inclusion of cases in the study and outcome after the first trimester (FT), second trimester (ST) and third trimester (TT) evaluations. (CNS—Central nervous system, US—Ultrasound, TOP—Termination of pregnancy).

**Figure 3 brainsci-13-00118-f003:**
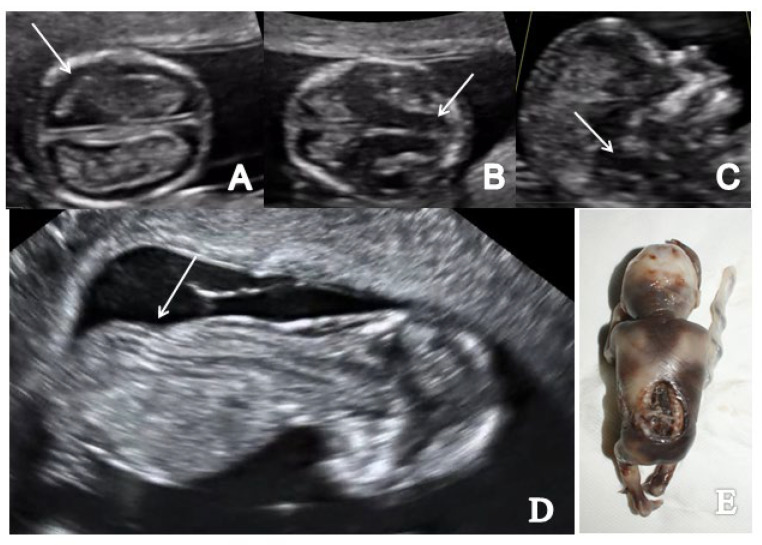
A case of OSB diagnosed at 12 weeks + 3 days (Case 1): (**A**) transverse view of enlarged choroid plexus (the ‘dried up’ brain sign, arrow); (**B**) displacement of mesencephalon and aqueduct of Sylvius and deformation against occipital bone (‘the crash sign’, arrow); (**C**) mid-sagittal view of the fetal face showing the displacement of BS, which appears thicker (arrow) and with an increased BS/BSOB ratio; (**D**) sagittal view by transvaginal approach demonstrating the abnormal aspect of the spine—kyphoscoliosis (arrow); (**E**) specimen presentation of OSB after medical TOP.

**Figure 4 brainsci-13-00118-f004:**
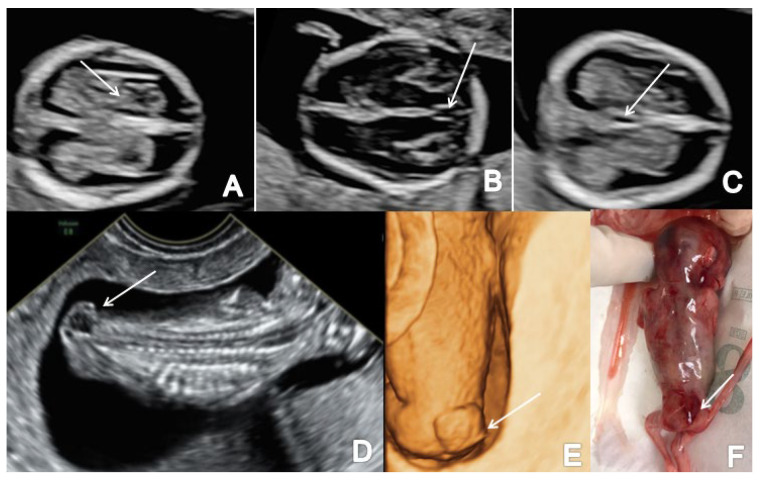
A case of OSB diagnosed at 13 weeks + 2 days (Case 3): (**A**) transverse view of the fetal head showing normal choroid plexus (arrow) and the filling of lateral cerebral ventricles; (**B**,**C**) further evaluation of the cerebral ventricular system, in transverse planes of the brain showing the aqueduct of Sylvius (**B**) and the third ventricle (**C**) indicated by arrows; (**D**) sagittal view by transvaginal approach demonstrating the defect (arrow); (**E**,**F**) specimen presentation after medical TOP.

**Figure 5 brainsci-13-00118-f005:**
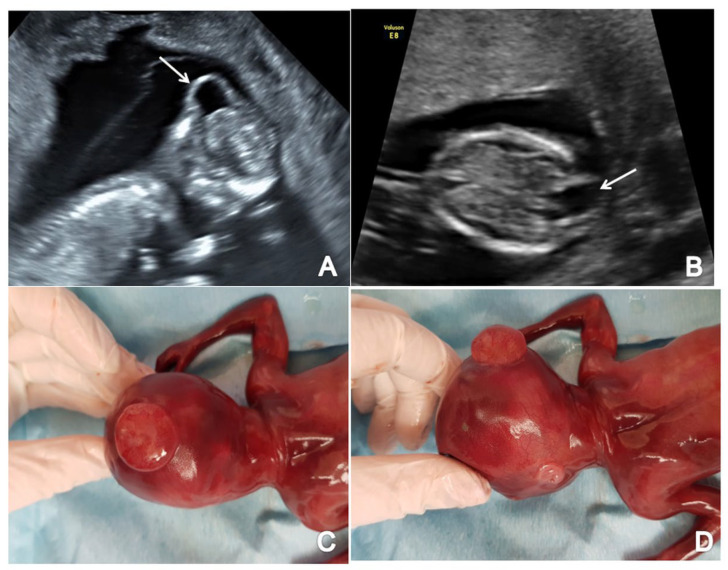
Ultrasound image of occipital cephalocele (the arrow marks the bony defect), diagnosed at 13 weeks + 6 days (Case 13): (**A**) sagittal view by transvaginal approach; (**B**) axial view by transabdominal approach; (**C**,**D**) specimen presentation after medical TOP.

**Figure 6 brainsci-13-00118-f006:**
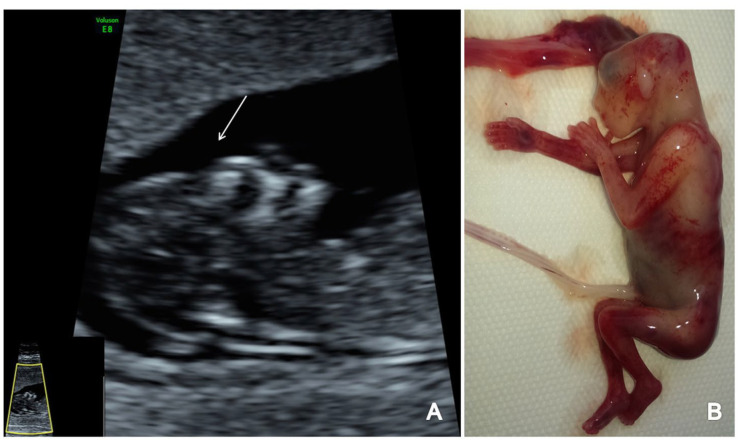
A case of anencephaly diagnosed at 12 weeks + 4 days weeks (Case7): (**A**) sagittal view of the absence of cranial bones ossification (arrow); (**B**) specimen presentation aspect after medical TOP.

**Figure 7 brainsci-13-00118-f007:**
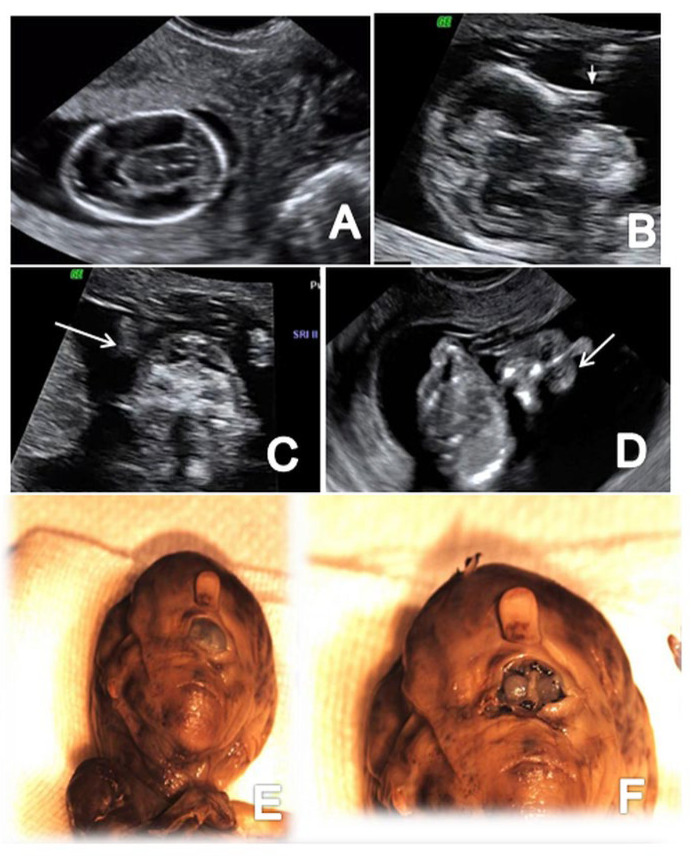
A case of alobar holoprosencephaly diagnosed at 12 weeks + 5 days (Case 9): (**A**) transverse view of the abnormal brain cavities; (**B**) sagittal view of the fetal head showing proboscis (arrow); (**C**) frontal view of the fetal face showing extreme hypotelorism (arrow); (**D**) frontal view of proboscis (arrow); (**E**,**F**) specimen presentation aspect after medical TOP confirming proboscis and hypertelorism.

**Figure 8 brainsci-13-00118-f008:**
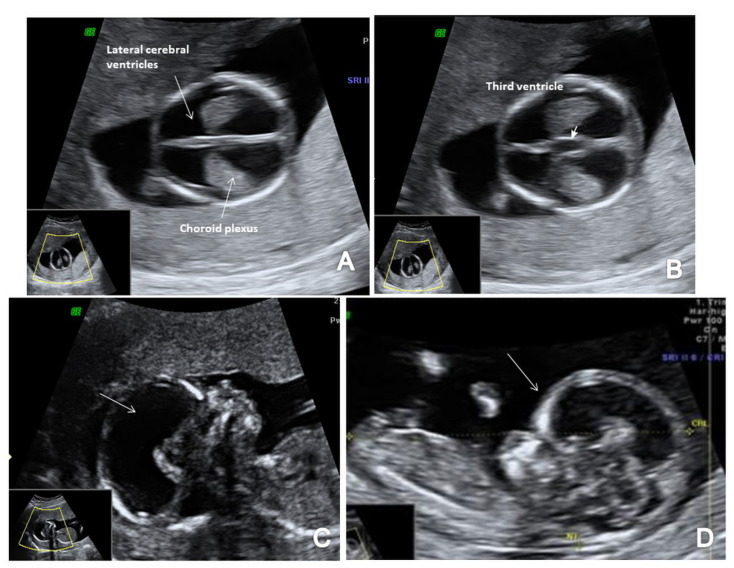
First-trimester ultrasound images of a fetus with hydrocephaly (Case 11): (**A**) enlarged cerebral ventricles with shrinked choroid plexus (axial view); (**B**) increased third ventricle (axial view); (**C**) abnormal sagittal view with increased forth ventricle and cisterna magna (arrow); (**D**) sagittal view with an important disproportion between head and body and frontal bossing (arrow).

**Table 1 brainsci-13-00118-t001:** The comparative satisfactory evaluation rate of the CNS features in the abnormal and normal group, reasons for not-satisfactory visualization by TAUS and the results after a reschedule and/or a TVUS.

	Identification of all CNS Features by TAUS	Associated Unfavorable Conditions Requiring a Reschedule/TVUS Examination 137/1943 Cases (7.05%)	CNS Features That Were Not Properly Identified at the Initial TAUS and Required TVUS or Re-Evaluation				
Abnormal group (major CNS anomalies + minor CNS US markers group)	22/26 cases (84.6%)	Fibroids 4 cases (2.9%) BMI > 24 32 cases (23.3%) unfavorable fetal position 11 cases (8%) retroverted uterus 11 cases (8%) Abdominal scar 23 cases (16.7%) Fibroids and abdominal scar 6 cases (4.3%) Unfavorable fetal position and increased BMI20 cases (14.5%) Increased BMI and abdominal scar 22 cases (16%)	The lateral ventricles and the choroid plexus	The third ventricle	The aqueduct of Sylvius	The fourth ventricle/Cisterna Magna/BS/BSOB	Spine and underlying skin
			2/4 cases (50%)	2/4 cases (50%)	1/4 cases (25%)	3/4 cases (75%)	1/4 cases (25%)
Normal group	1784/1917 cases (93%)		13/133 cases (9.7%)	37/133 cases (27.8%)	52/133 cases (39%)	71/133 cases (53.3%)	88/133 cases (66.1%)

TA-transabdominal, US- ultrasound, BMI- body mass index, TV-transvaginal.

**Table 2 brainsci-13-00118-t002:** Abnormal US features, karyotype and outcome of cases with FT CNS anomalies.

No.	GA	Ultrasound Findings Respecting the Scanning Protocol							Diagnosis	Karyotype	Outcome
		The Lateral Ventricles and the Choroid Plexus	The Third Ventricle	The Aqueduct of Sylvius	The Fourth Ventricle	Cisterna Magna	BS/BSOB	The Spine			
1.	12 w + 3 d	+	+	+	+	+	+	+	Open Spina Bifida	46XY	1st trimester TOP
2.	12 w + 5 d	−	+	+	+	+	+	+	Open Spina Bifida	Trisomy 18	1st trimester TOP (surgical)
3.	13 w + 2 d	−	−	−	−	−	−	+	Open Spina Bifida	46XX	1st trimester TOP
4.	13 w + 1 d	−	−	−	−	−	−	+	Closed Spina Bifida	46XX	Early 2st trimester TOP
5	12 w + 0 d	+	+	+	+	+	+	−	Anencephaly	unknown	Miscarriage
6.	12 w + 1 d	+	+	+	+	+	+	−	Anencephaly	unknown	1st trimester TOP
7.	12 w + 4 d	+	+	+	+	+	+	−	Anencephaly	46XX	1st trimester TOP
8.	12 w + 3 d	+	+	+	+	+	+	−	Holoprosencephaly	unknown	1st trimester TOP (surgical)
9.	12 w + 5 d	+	+	+	+	+	+	−	Holoprosencephaly	unknown	1st trimester TOP
10.	13 w + 1 d	+	+	−	+	+	+	−	Holoprosencephaly	Trisomy 18	1st trimester TOP
11.	13 w + 5 d	+	+	+	+	+	+	−	Hydrocephaly	Trisomy 21	Early 2st trimester TOP
12.	13 w + 4 d	−	+	+	−	+	−	−	Cephalocele	Trisomy 21	1st trimester TOP
13.	13 w + 6 d	−	+	+	−	+	−	−	Cephalocele	unknown	1st trimester TOP
14	12 w + 6 d	−	−	−	+	+	+	−	Dandy-Walker malformation	unknown	1st trimester TOP
15	13 w + 1 d	−	−	−	+	+	+	−	Dandy-Walker malformation	Trisomy 18	2st trimester TOP

GA (Gestational age at diagnosis), TOP (Termination of pregnancy), + (Abnormal ultrasonographic finding), − (normal ultrasonographic finding).

## Data Availability

All data presented here is available from the authors, upon reasonable request.

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
