# Peer review of "First Trimester Ultrasound Detection of Fetal Central Nervous System Anomalies"

_brainsci, 2023, doi:10.3390/brainsci13010118_

Round 1

Reviewer 1 Report

This is a well-written article that makes clear which brain and spine abnormalities we could detect and should search for in the first trimester scan and the detection rate that should be discussed with the mother/couple. Strengths of the study is the prospective design and the large number of cases included. The introduction provides sufficient background from literature review. Inclusion criteria are clearly presented. Research design is appropriate and the statistical analysis is adequate. Conclusions are supported by the results and discussion is to the point. Illustrations and graphs also make the paper very educative for young and advanced sonographers. I believe that the article is of great scientific value.

Author Response

December 2022

Roxana Cristina Drăgușin, MD, PhD

Department of Obstetrics and Gynecology

University of Medicine and Pharmacy of Craiova

Department of Obstetrics and Gynecology II, Emergency University Hospital Craiova

2-4 Petru RareÈ™ Street, 200349, Craiova, Romania

Phone: +40769251576

e-mail: roxy_dimieru@yahoo.com

To the Editor of Brain Sciences

           We are grateful for the reviewer’s commitment to help in reporting accurately the results and for his/hers efforts to make the most of our work.

We confirm that neither the manuscript nor any parts of its content are currently under consideration or published in another journal.

All authors have approved the manuscript and agree with its submission to Brain Sciences.

            We want to inform you about the latest changes performed after the first revision. The latest version of the manuscript has been uploaded on the journal's page. We used "track changes" for any change to the manuscript.

General improvements of the manuscript:

  • We had the manuscript revised by a native English-speaking colleague who improved the English language and style;
  • We wrote all references using the MDPI Reference List and Citations Style Guide.
  • Line 32 : we changed the phrase into : ”The cases were also followed during the second and third trimester of pregnancy and at delivery.”
  • Line 41: we replaced ”not present” with ”absent”.
  • Line 44: we removed the word ”vast”.
  • Line 66: we changed the phrase into : ”Early detection of major CNS diseases offers the possibility for a safer termination of pregnancy with less emotional stress and less economic costs”.
  • Line 79: we added the Ethical Approval Code : ”(No 29/20.03.2018)”.
  • Line 101: we replaced ”pregnant women” with ”study group”.
  • Line 128: we changed the phrase into : ”After termination of pregnancy, autopsy of the fetuses with major FT CNS anomalies was offered”.
  • Line 166: we added ”of the pregnant uterus”.
  • Line 210: we deleted ”and one case presented an occipital meningocele”.
  • Line 258: we replaced ”similar to other reports” with ”previously reported”.
  • Line 277: we replaced ”similar to other published papers” with ”as mentioned before”.
  • Line 375: we deleted ”Figure 6” as it was previously mentioned in the section called Results.
  • Line 393: we replaced the reference ”67” with ”68”.
  • Line 400: we deleted ”associated other CNS anomalies”.
  • Line 411: we deleted ”Figure 7” as it was previously mentioned in the section called Results.
  • Line 487: we added ”Conceptualization, M.I; Methodology, L.Z, A.I-O, M.S.V and L.A.D; Formal analysis, A.C.C; Investigation, D.R.U, R.G.C, C.M, C.L.P, M.C.C and O.C.S; Resources, I.S, F.B and A.N.D; Writing – original draft, R.C.D; Writing – review & editing, D.G.I. All authors have read and agreed to the published version of the manuscript.”
  • Line 496: we added the Ethical Approval Code ”(No 29/20.03.2018).”

Thank you.

Sincerely yours,

Roxana Cristina Drăgușin, MD, PhD.

Reviewer 2 Report

The paper entitled „FIRST TRIMESTER ULTRASOUND DETECTION OF FETAL CENTRAL NERVOUS SYSTEM ANOMALIES” (brainsci-2113701) is very interesting. The presented research combines theoretical and practical observations, complementing the knowledge on the neuroanatomy of the central nervous system in early prenatal development.

The study showed that early prenatal diagnosis (first trimester of pregnancy) is a very important factor in detecting CNS defects. The authors carried out extensive observations on 1943 fetuses, in which frequent and rare defects were detected. In the presented studies, the authors have shown that a detailed assessment of the CNS in the early period of pregnancy is possible and has a high potential for effective detection of large CNS anomalies with an accuracy of 99.4%. Detection of CNS malformations in FT allows for faster and safe termination of pregnancy. This study also confirms previous reports that combined intracranial features such as CM blurring, a BS/BSOB ratio greater than 1, and the presence of a "crash" sign are highly effective in detecting OSB in FT. The great value of this publication is a broader view of known cases of CNS defects, but with greater accuracy. The precision, accuracy and thoroughness of ultrasound examinations allowed us to notice new details, "reference points", which may constitute important premises for the emergence of a potential CNS defect in the later stage of prenatal development. These are very important diagnostic observations.

However, there are following some points for corrections prior to publication:

Please check the correctness of the English language

Please correct the „Literature section” according to the MDPI Reference List and Citations Style Guide

Please check the reference [68], I did not find a citation in the text

Author Response

December 2022

Roxana Cristina Drăgușin, MD, PhD

Department of Obstetrics and Gynecology

University of Medicine and Pharmacy of Craiova

Department of Obstetrics and Gynecology II, Emergency University Hospital Craiova

2-4 Petru RareÈ™ Street, 200349, Craiova, Romania

Phone: +40769251576

e-mail: roxy_dimieru@yahoo.com

To the Editor of Brain Sciences

           We are grateful for the reviewer’s commitment to help in reporting accurately the results and for his/hers efforts to make the most of our work.

We confirm that neither the manuscript nor any parts of its content are currently under consideration or published in another journal.

All authors have approved the manuscript and agree with its submission to Brain Sciences.

            We want to inform you about the latest changes performed after the first revision. The latest version of the manuscript has been uploaded on the journal's page. We used "track changes" for any change to the manuscript.

General improvements of the manuscript:

  • We had the manuscript revised by a native English-speaking colleague who improved the English language and style;
  • We wrote all references using the MDPI Reference List and Citations Style Guide.
  • Line 32 : we changed the phrase into : ”The cases were also followed during the second and third trimester of pregnancy and at delivery.”
  • Line 41: we replaced ”not present” with ”absent”.
  • Line 44: we removed the word ”vast”.
  • Line 66: we changed the phrase into : ”Early detection of major CNS diseases offers the possibility for a safer termination of pregnancy with less emotional stress and less economic costs”.
  • Line 79: we added the Ethical Approval Code : ”(No 29/20.03.2018)”.
  • Line 101: we replaced ”pregnant women” with ”study group”.
  • Line 128: we changed the phrase into : ”After termination of pregnancy, autopsy of the fetuses with major FT CNS anomalies was offered”.
  • Line 166: we added ”of the pregnant uterus”.
  • Line 210: we deleted ”and one case presented an occipital meningocele”.
  • Line 258: we replaced ”similar to other reports” with ”previously reported”.
  • Line 277: we replaced ”similar to other published papers” with ”as mentioned before”.
  • Line 375: we deleted ”Figure 6” as it was previously mentioned in the section called Results.
  • Line 393: we replaced the reference ”67” with ”68”.
  • Line 400: we deleted ”associated other CNS anomalies”.
  • Line 411: we deleted ”Figure 7” as it was previously mentioned in the section called Results.
  • Line 487: we added ”Conceptualization, M.I; Methodology, L.Z, A.I-O, M.S.V and L.A.D; Formal analysis, A.C.C; Investigation, D.R.U, R.G.C, C.M, C.L.P, M.C.C and O.C.S; Resources, I.S, F.B and A.N.D; Writing – original draft, R.C.D; Writing – review & editing, D.G.I. All authors have read and agreed to the published version of the manuscript.”
  • Line 496: we added the Ethical Approval Code ”(No 29/20.03.2018).”

Thank you.

Sincerely yours,

 Roxana Cristina DrăguÈ™in, MD, PhD.
